# Application of machine learning in predicting consumer behavior and precision marketing

Jin Lin *

College of New Media, Yango University, Fuzhou City, China

* ctmr99@163.com

## Abstract

with the intensification of market competition and the complexity of consumer behavior, enterprises are faced with the challenge of how to accurately identify potential customers and improve user conversion rate. This paper aims to study the application of machine learning in consumer behavior prediction and precision marketing. Four models, namely support vector machine (SVM), extreme gradient boosting (XGBoost), categorical boosting (CatBoost), and backpropagation artificial neural network (BPANN), are mainly used to predict consumers' purchase intention, and the performance of these models in different scenarios is verified through experiments. The results show that CatBoost and XGBoost have the best prediction results when dealing with complex features and large-scale data, F1 scores are 0.93 and 0.92 respectively, and CatBoost's ROC AUC reaches the highest value of 0.985. while SVM has an advantage in accuracy rate, but slightly underperformance when dealing with large-scale data. Through feature importance analysis, we identify the significant impact of page views, residence time and other features on purchasing behavior. Based on the model prediction results, this paper proposes the specific application of optimization marketing strategies such as recommendation system, dynamic pricing and personalized advertising. Future research could improve the predictive power of the model by introducing more kinds of unstructured data, such as consumer reviews, images, videos, and social media data. In addition, the use of deep learning models, such as Transformers or Self-Attention Mechanisms, can better capture complex patterns in long time series data.

## 1 Introduction

With the intensification of market competition, consumer behavior becomes more and more complex. Traditional marketing methods often struggle to cope with this complexity [1,2]. Traditional marketing models are often too rigid, based on historical data and static analysis, to respond to rapidly changing market demands. For example, the 4P model (product, price, promotion, channel) cannot effectively handle

**Data availability statement:** The data used in this study is from the publicly available dataset Online Shoppers Purchasing Intention Dataset (version: 2016), which comes from the UCI Machine Learning Library.
link: https://archive.ics.uci.edu/ml/datasets/ Online+Shoppers+Purchasing+Intention+Dataset

**Funding:** The author(s) received no specific funding for this work.

**Competing interests:** The authors have declared that no competing interests exist.

multidimensional data on modern consumer behavior, especially in the context of large-scale, real-time data streams Consumer behavior is not only influenced by multiple factors such as culture, society and psychology, but also involves complex variables such as information overload, personalized demand and purchase motivation. With the intensification of market competition, enterprises are in urgent need of innovative means to accurately identify potential consumers and improve customer retention, so as to increase market share [3,4]. Precision marketing, as an effective way to attract and maintain customers through personalized communication, is becoming one of the core strategies of marketing[5].

The research of consumer behavior has always been the core issue in the field of marketing. In recent years, researchers have made use of behavioral economic models and psychological theories to deeply explore the decision-making process of consumers [6]. These models help enterprises better understand consumers' purchase motivation, cognitive response and emotional attitude, and provide theoretical basis for the design of marketing strategies [7]. However, with the rapid development of digital technology, consumer behavior has gradually migrated from offline to online, and this change in behavior pattern has brought new challenges and opportunities for researchers and enterprises. Through social media, mobile devices, and e-commerce platforms, businesses have access to vast amounts of consumer behavior data, which provides a valuable resource for predicting consumer behavior.

In recent years, with the development of big data technology and artificial intelligence, the application of machine learning in marketing has shown great potential [8]. Machine learning algorithms can process vast amounts of consumer data to identify hidden patterns and trends, which in turn provide insights to businesses to help optimize marketing decisions. In precision marketing, machine learning can accurately predict consumers' future behavior and develop personalized recommendation strategies by analyzing consumers' historical behavior and interaction data [9,10]. Machine learning models such as SVM, XGBoost, CatBoost, and BPANN have proven their superiority in many fields, especially when dealing with large data and complex features. Studies have shown that the accuracy and scalability of XGBoost and CatBoost on large-scale data sets are significantly better than traditional algorithms [11], while SVM is ahead in accuracy and is especially suitable for the classification of high-dimensional features [12]. Zhao [13] reviewed the application of machine learning and artificial intelligence in precision marketing. Deep learning and natural language processing models can improve prediction accuracy by about 25% in consumer sentiment analysis. The article points out that CatBoost and XGBoost have become the mainstream models of e-commerce recommendation systems, which can effectively improve the conversion rate of personalized ads, up to 30%. Luo [14] explored the application of machine learning on e-commerce platforms, especially customer behavior prediction and personalized recommendations. Fang [15] reviews the latest applications of machine learning and artificial intelligence in marketing, especially in retail and e-commerce. The application of deep learning and reinforcement learning techniques in personalized recommendation systems has increased customer

engagement by about 20%. Bucklin et al. [16] pointed out that machine learning can promote the automation and precision of marketing decisions by analyzing complex multidimensional data. Sahoo et al. [17] believe that machine learning can help enterprises automate complex marketing tasks, optimize resource allocation and improve return on marketing investment. In addition, using machine learning algorithms for consumer segmentation, companies are able to accurately identify high-value customer segments based on different buying preferences and behavior patterns, and tailor personalized marketing parties to these groups. Recommendation system is one of the most typical applications of machine learning in the field of marketing. By analyzing user behavior patterns, the recommendation system can provide personalized product recommendations to consumers, significantly improving customer conversion rates and satisfaction. Amazon and Netflix have significantly increased conversion rates through their recommendation systems, with Netflix's personalized recommendation system increasing engagement by more than 30% [18]. In terms of market segmentation, through clustering algorithms, machine learning can divide consumers with different characteristics into multiple groups, enabling enterprises to develop targeted marketing strategies for each market segment, thus improving the effect of marketing [19–23]. In addition to recommendation systems, machine learning is also widely used in dynamic pricing, customer segmentation and advertising [24,25]. For example, in e-commerce platforms, machine learning algorithms can analyze consumers' purchase history and browsing behavior in real time, so as to adjust product pricing and attract more potential customers [26].

However, despite numerous studies exploring the application of different machine learning models in this field, the comprehensive advantages of models such as CatBoost and XGBoost in e-commerce data processing have not been fully explored. By comparing SVM, XGBoost, CatBoost and BPANN four mainstream machine learning models, this study explores their performance in large-scale and complex e-commerce data to fill this gap. The accuracy and generalization ability of these models in processing e-commerce behavior data are comprehensively evaluated, and specific insights are provided on how to select appropriate algorithms.

## 2 Methods and materials

In this study, the modeling process consists of several steps, including data collection, data preprocessing (including feature selection, etc.), model selection and training (including SVM, XGBoost, CatBoost, BPANN models), model evaluation and optimization.

### 2.1 Data collection

The data set used in this paper is the Online Shoppers Purchasing Intention Dataset (Version: 2016), which comes from the UCI machine learning library and contains 12,330 independent user sessions, which records the user's behavior and environment information on the e-commerce platform. The data covers user engagement metrics such as page visits, stay times, bounce rates, exit rates, page value, etc. All data are anonymous user sessions over a year, avoiding bias for specific events or dates. The dataset contains 10 numerical types and 8 classification features, and the target variable is Revenue, which represents whether the user completes the purchase. The data set provides a good feature basis for user purchase behavior prediction and ensures the stability and effectiveness of the model.

### 2.2 Modeling algorithms

(1) Support Vector Machine

Support Vector Machine (SVM) distinguishes different classes by finding an optimal hyperplane in the feature space (as shown in Fig 1 ( a ) ), focusing on support vectors—data points closest to the hyperplane. By maximizing the margin between these points and the hyperplane, the model achieves better generalization performance[27]. In this study, we chose the SVM model because it has good generalization ability and is especially suitable for high-dimensional data and

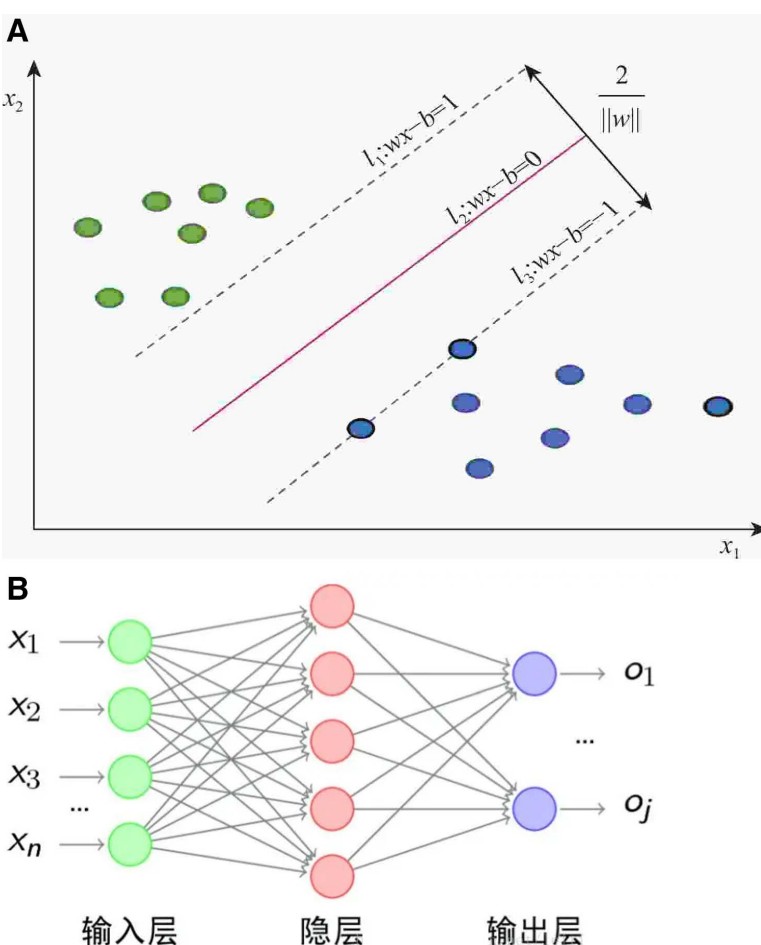

**Fig 1. Model diagram. (a) SWM.(b) BPANN.**

classification tasks [28]. finds the optimal hyperplane for classification, uses RBF kernel function, and makes full use of support vector to improve the generalization ability of the model.

(2) XGBoost

XGBoost is a refined gradient boosting algorithm built on the GBDT framework, incorporating techniques to enhance performance and prevent overfitting, such as a regularization term in the loss function. The complexity of the model is controlled by the number of leaf nodes (T) and their weights (w):

$$L = \sum_{i=1}^{n} L\left(y_i, \hat{y}_i\right) + \sum_{k=1}^{K}\left(\gamma T_k + \frac{1}{2}\lambda \left\|W_k\right\|^2\right)$$

(1)

Shrinkage reduces the impact of each tree on the final model, improving convergence and generalization. XGBoost optimizes the complexity of the model by integrating learning and regularizing terms [29]. In large-scale data processing and complex feature analysis, XGBoost performs very well, and its processing speed and scalability are better than traditional

models [30]. We chose XGBoost because its efficient parallelization and automated feature selection methods make it ideal for data mining tasks in the e-commerce field.

(3) CatBoost

CatBoost, short for "Categorical Boosting," uses symmetric decision trees and the Ordered Target Statistic method, which calculates statistics based on historical data for each sample. A prior value reduces noise from low-frequency categories, allowing effective handling of categorical features:

$$x_{i,k} = \frac{\sum_{j=1}^{p-1} [x_{\sigma j,k} = x_{\sigma p,k}] \cdot Y_j + a \cdot p}{\sum_{j=1}^{p-1} [x_{\sigma j,k} = x_{\sigma p,k}] + a}$$

(2)

CatBoost, on the other hand, uses the ordered target statistics method of class features, which has advantages when dealing with high-dimensional class features. CatBoost's accuracy and ability to handle complex categorical data make it highly effective for large datasets, outperforming other GBDT frameworks [31]. Previous studies have demonstrated the advantages of XGBoost and CatBoost in large-scale data processing and feature complexity analysis. Based on these studies, this paper further explores their application in e-commerce data.

(4) BPANN

Backpropagation Artificial Neural Network (BPANN) models complex nonlinear relationships through gradient descent (as shown in Fig 1 ( b ) ). The training process includes forward propagation, error calculation, backpropagation, and weight updating, iterating until the prediction error is minimized or the preset iteration limit is reached [32,33]. BPANN has been widely used in many fields, especially in the analysis of consumer behavior, which can effectively capture the non-linear decision-making mode of consumers [34]. Although its computational complexity is high, it still has important application value in dealing with complex nonlinear problems.

## 2.3 Data preprocessing and feature engineering

(1) Data preprocessing

First, categorical variables including month (Mth), visitor type (VT), weekend or not (Wknd) are encoded and converted to numeric data using LabelEncoder. For the rest of the numerical features, StandardScaler is used to standardize the range of values. Using IsolationForest, the contamination rate was set to 0.01, and extreme outliers were identified and removed to improve model stability and robustness. SMOTE method was used to up-sample the target variable Rev to deal with the class imbalance in the dataset. SMOTE method was used to deal with the category imbalance, SMOTE effectively avoids model bias due to sample imbalance by synthesizing new minority class samples to balance the dataset.

(2) Feature engineering

Variance filtering with a threshold of = 0.05 is performed to remove non-normal distribution data with skewness greater than 1, because these features have limited predictive contributions to the model. The mutual information method with a threshold of 0.05 is designed to retain the features that are strongly correlated with the target variable and remove the redundant information that is unrelated to the target variable. Finally, the features with absolute correlation score greater than 0.7 between the two features were deleted through the Pearson correlation coefficient. The features with smaller Rev correlation coefficient are preferentially deleted. Fig 2 is a Pearson heat map, showing the correlation coefficients among each characteristic parameter.

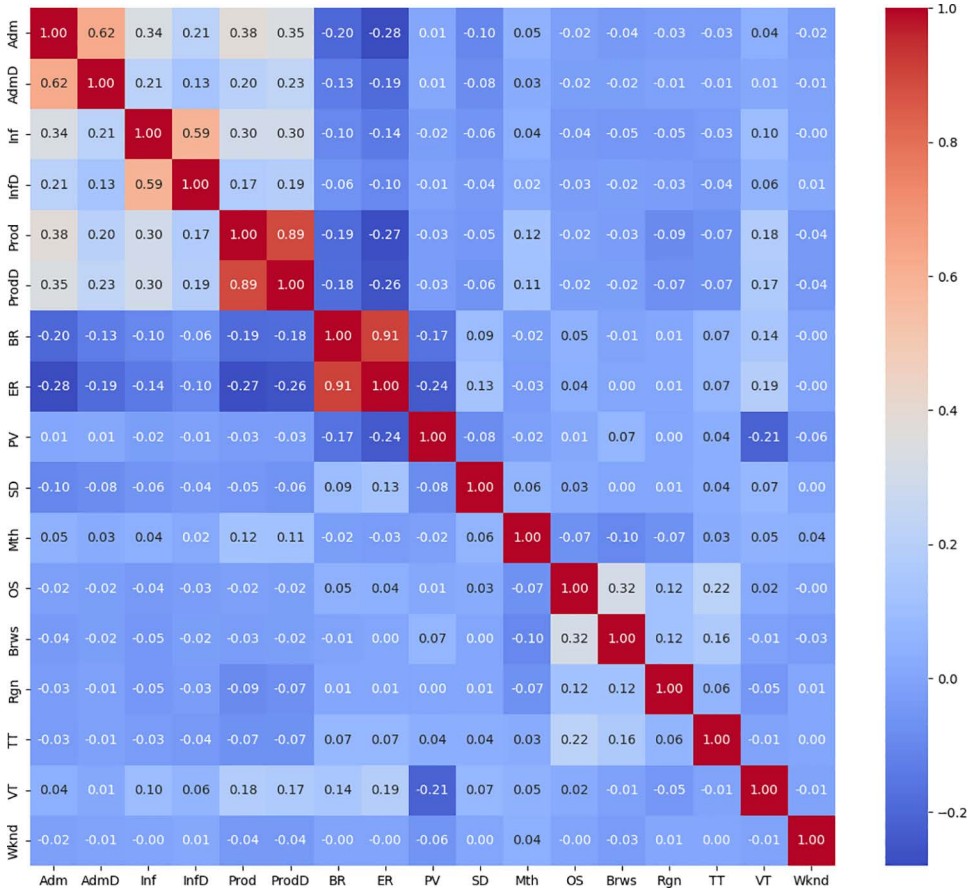

**Fig 2. Pearson correlation coefficient matrix diagram.**

## 2.4 Model training and optimization

After feature screening, the model was initially trained and optimized. SVM, XGBoost, CatBoost and BPANN nonlinear classification algorithms were used for training based on the amount of data and its data features. The training set and the test set were divided by 0.20 test_size, and 50 random numbers were set by a cyclic algorithm for data distribution. In order to comprehensively evaluate the performance of the model, we used ROC AUC, accuracy and recall rate. The ROC AUC measures the model's performance at different thresholds, the accuracy rate reflects the proportion of models that predict correctly, while the recall rate represents the model's ability to identify positive samples. These indicators together evaluate the effectiveness of the model in different application scenarios. In the process of model optimization, grid search and random search are used to optimize the hyperparameters. The hyperparameters were optimized using Grid Search and Random Search. Specific hyperparameters, such as the C value of SVM and max_depth of XGBoost, are listed in Table 1. RBF is used to build SVM, and the integrated XGBoost and CatBoost algorithms optimize parameters such as max_depth through random search to improve the accuracy of the model and the ability to process category features. In the optimization of BPANN, the number of neural network layers and the number of nodes in each layer are adjusted, and the early stop method is used to prevent overfitting. Hyperparameter optimization significantly improves the performance of the model, and the accuracy and recall rate of the model are significantly improved. Fig 3 shows the confusion matrix after hyperparameter optimization of CatBoost and SVM.

**Table 1. Model parameter values.**

| argument | SVM | XGBoost | CatBoost | BPANN |
|---|---|---|---|---|
| epsilon | 0.019 | | | |
| gamma | 0.120 | | | |
| C | 50.12 | | | |
| iterations | | | 245 | |
| learning_rate | | 0.11 | 0.09 | |
| max_depth | | 4 | 8 | |
| subsample | | 0.8 | | |
| hidden_layers | | | | 3 |
| epochs | | | | 50 |

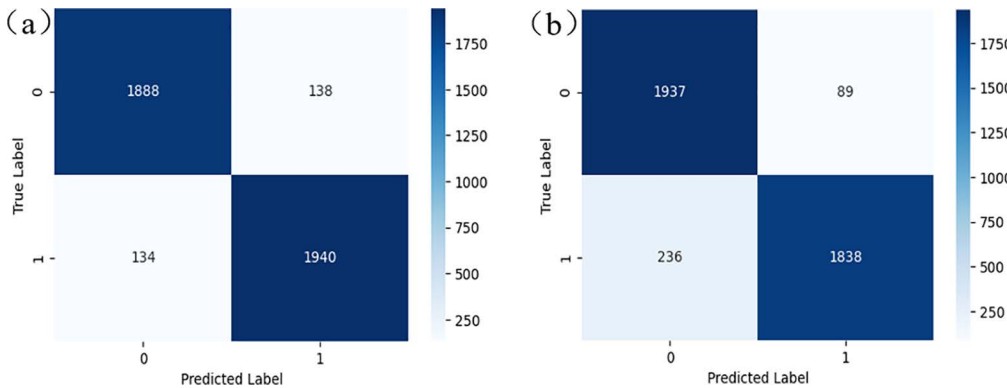

**Fig 3. Confusion matrix. (a) CatBoost. (b) SWM.**

## 3  Results and discussion

### 3.1  Model performance evaluation and comparison

The fitting effects of the four models on the test set and the performance comparison of the model test set are shown in in Table 2 and Fig 4, respectively. CatBoost demonstrates the best overall performance across all metrics, with an accuracy rate of 93.4%, a recall rate of 93.5%, and the highest ROC AUC of 0.985. This model excels in handling large-scale and complex data tasks, making it particularly suitable for e-commerce data processing, where complex feature interactions and noise are prevalent. XGBoost, similar to CatBoost, performs well across all indicators, with a precision of 93.5%, a recall of 92.5%, and a ROC AUC of 0.984. XGBoost's balanced performance and strong generalization ability make it another excellent choice for handling large datasets with complex features. While SVM achieves the highest precision (95.4%), it has a relatively low recall rate (88.6%) and performs well in minimizing false positives. However, its ability to identify positive samples is slightly weaker than that of CatBoost and XGBoost, which makes it more suitable for tasks where high accuracy is crucial, but not necessarily at the expense of recall. The BPANN model, though the least perform-ing model in this comparison, still provides a balanced outcome with a precision of 90.1%, a recall of 90.2%, and a ROC AUC of 0.955. BPANN is particularly suited for tasks involving complex, nonlinear relationships, although its performance lags behind other models in large-scale tasks.

By comparing the performance of SVM, XGBoost, CatBoost, and BPANN on the test set, we identified the advantages of each model in a specific context. CatBoost performed best, with a high ROC AUC value (0.985) on large-scale data

**Table 2. The fitting effect of four models on the test set.**

| Models | Precision | Recall | ROC AUC |
|---|---|---|---|
| SVM | 0.954 | 0.886 | 0.977 |
| XGBoost | 0.935 | 0.925 | 0.984 |
| CatBoost | 0.934 | 0.935 | 0.985 |
| BPANN | 0.901 | 0.902 | 0.955 |

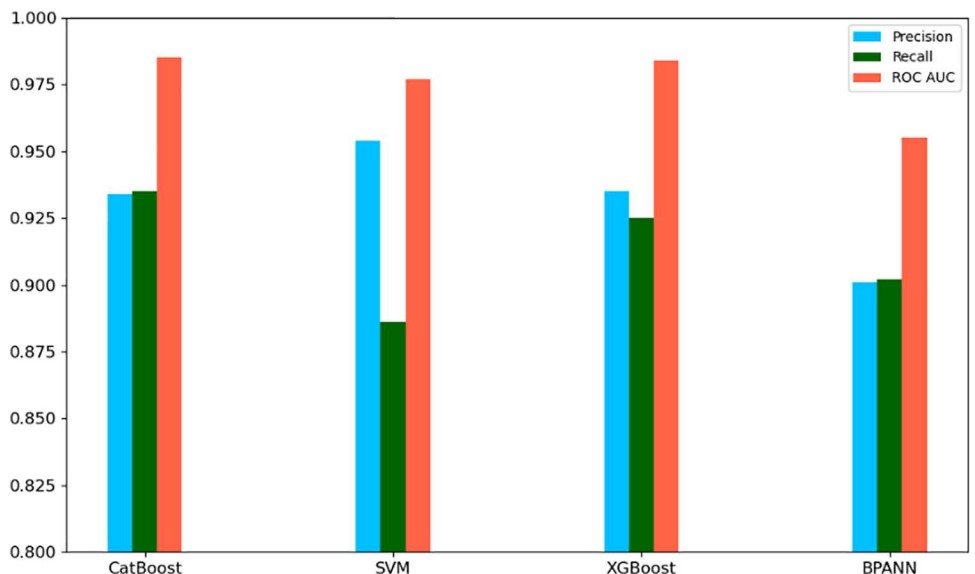

**Fig 4. Model test set performance comparison.**

and complex feature processing confirming its effectiveness in e-commerce data. CatBoost and XGBoost are the optimal choices, which are outstanding in the consumer behavior prediction and precision marketing scenarios of e-commerce. Both of them have excellent classification performance and anti-noise ability when dealing with large-scale data and high-dimensional features. Although SVM has obvious advantages in accuracy rate, it is more suitable for small-scale and high-dimensional tasks due to its low recall rate and efficiency in processing large-scale data. Although BPANN is slightly inferior in classification tasks, it has certain application value in tasks that need to deal with complex nonlinear relationships.

### 3.5 Feature importance analysis

The analysis relies on the importance score of each feature calculated by CatBoost during training to reflect the relative contribution of the feature to the model's predictive power. By analyzing the importance of features, we can identify which user behaviors or environmental characteristics have the most influence on purchasing behavior

(1) Visual display of feature importance

As shown in Fig 5, the importance distribution of each feature to purchase behavior prediction can be clearly seen through the feature importance analysis of CatBoost model. Adm and PV were the two features that contributed the most to the predicted results, indicating that they played a crucial role in influencing user buying behavior. Other features, such as Mth and Prod, also have an impact on users' purchase decisions to some extent.

 

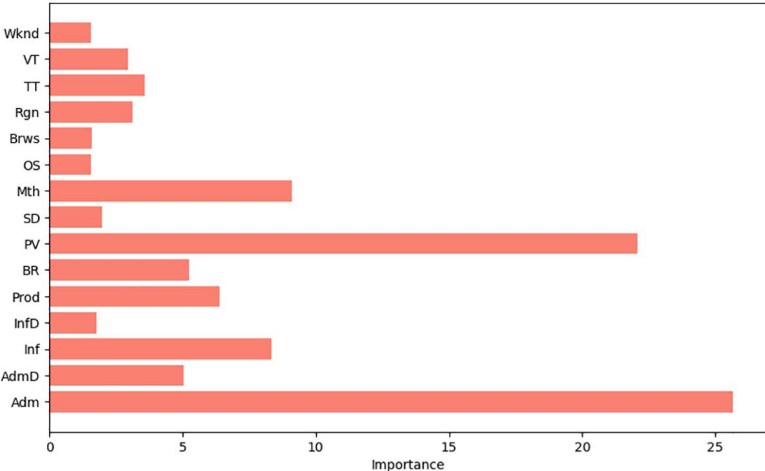

**Fig 5. CatBoost feature importance ranking of models.**

(2) Analyze the most influential characteristics

The more Product pages users visit on the platform, it may indicate that users have a higher degree of trust in the platform, and the possibility of purchase will also increase. The increase in page views means that the user's interest in the platform product is growing, so the likelihood of purchase is increased; Different months may correspond to specific marketing campaigns or shopping holidays, so this feature also contributes significantly to predicting users' purchase behavior; The more pages that users visit directly related to the product, the higher their interest in the product, and the greater the likelihood of purchase.

(3) How to use feature importance to optimize marketing strategy

Improve user engagement: For features that have a large impact, such as page views and residence time, enterprises can extend the residence time of users on the platform by optimizing page layout, improving page loading speed, improving user experience, and so on, thereby increasing the possibility of purchase.

Personalized recommendation: The personalized recommendation system uses machine learning models (such as collaborative filtering, deep learning models, etc.) to provide customized recommendations by analyzing users' historical behavior, interest preferences, and interactions with products. According to the categories and related pages of the products viewed by the user, the enterprise can recommend relevant products to the user through the personalized recommendation system. By analyzing user visits to product-related pages, you can significantly improve the accuracy and conversion rate of recommended products. Using machine learning algorithms, Netflix has increased user engagement and subscription rates through a personalized recommendation system, which is able to push relevant content based on a user's viewing history and interests, significantly increasing user conversion rates.

Reduce bounce rate: For pages with high bounce rate, enterprises can reduce user bounce behavior by redesigning the page content or guiding users to more relevant content, thereby improving user purchase intention.

Optimize advertising: Feature importance analysis can also help enterprises determine the characteristics of high purchase intention users, so as to focus advertising resources, optimize advertising accuracy, and improve the use of marketing resources. Companies such as Amazon and Facebook use machine learning to deliver targeted advertising based on user behavior prediction, improving AD effectiveness and ROI. Based on a user's browsing history, shopping preferences, and interest tags, advertisers can deliver relevant ads that significantly increase conversion rates and user engagement.

### 3.6 The optimization of marketing strategy

Based on the predictive model of consumer behavior, marketing strategies can be optimized in many ways. The personalized recommendation system uses machine learning models (such as collaborative filtering, deep learning models, etc.) to provide customized recommendations by analyzing users' historical behavior, interest preferences, and interactions with products. For example, based on the user's browsing history, the system can push similar products or items related to the user's interests, thus improving the relevance and accuracy of recommendations; Dynamic pricing Automatically adjusts the price of goods or services by analyzing market demand, consumer behavior, and competitor pricing in real time. Machine learning algorithms can help businesses predict price movements and make adjustments based on factors such as users' purchasing power, fluctuations in demand, seasonal changes, and more. At the same time, it can identify high loyalty users, and launch targeted membership programs and preferential activities to enhance brand stickiness. In addition, predictive models can help companies identify potential churn users, intervene in a timely manner through personalized offers and services, reduce churn, and ultimately increase user conversion and loyalty.

### 3.7 Conclusion and future scope

Through the application of SVM, XGBoost, CatBoost, and BPANN machine learning models in consumer behavior prediction, this paper successfully verifies the effectiveness of these models, particularly in the context of precision marketing within the e-commerce industry. The study aligns with the initial research questions and hypotheses that aimed to enhance consumer behavior prediction and optimize precision marketing strategies through machine learning. The findings confirm that CatBoost and XGBoost, due to their ability to handle large-scale data and complex features, are the best-performing models in terms of prediction accuracy and generalization ability, making them especially suitable for e-commerce applications. This directly aligns with the study's objective of improving user conversion rates and the overall effectiveness of precision marketing. While SVM excels at high-precision tasks, its lower efficiency in handling large amounts of data makes it more suitable for small-scale and high-dimensional tasks, consistent with our hypothesis that SVM would be effective for tasks demanding high accuracy but less scalability. BPANN, on the other hand, provides valuable insights when dealing with nonlinear relationships but has some limitations in large-scale data processing, as anticipated in our research.

This study not only provides accurate consumer behavior prediction methods for e-commerce platforms but also demonstrates significant value in practical applications such as personalized recommendations, dynamic pricing, and precision marketing. The results show that by accurately predicting consumer behavior, businesses can improve the utilization efficiency of marketing resources, enhance user conversion rates, and gain a competitive edge in the increasingly fierce market competition, directly supporting the initial research objectives.

Additionally, the research provides valuable insights that can be applied to other industries, such as finance and healthcare, promoting the adoption of machine learning methods across various fields of society. The potential for machine learning in precision marketing extends beyond e-commerce, and future research can leverage more diverse data types, such as social media behavior, geolocation, and other unstructured data, to further improve model performance. For example, integrating social media behavior data can capture real-time user sentiment and intent, enhancing prediction accuracy by identifying consumer preferences and behavioral patterns that are not captured through traditional data sources. Incorporating deep learning and reinforcement learning techniques is a promising avenue for improving user behavior prediction. By integrating these advanced techniques, the model can learn more complex patterns, such as temporal behavior changes and sequential decision-making. However, challenges such as increased computational complexity, training time, and data requirements need to be addressed. In the context of reinforcement learning, the model would require real-time feedback to adjust its predictions and optimize marketing strategies dynamically.

This study uses static data sets and does not adequately capture dynamic changes in consumer behavior. This can affect how the model performs in real-world applications, especially in the face of rapidly changing market conditions. Future research could consider introducing real-time data to adapt the model to rapid changes in consumer behavior by dynamically updating it. Future work should address the limitations and gaps identified in this study. Model performance refinements can focus on overcoming challenges such as overfitting or improving scalability for even larger datasets. Additionally, alternative strategies, such as using ensemble models or feature selection techniques, could be explored to enhance model robustness and interpretability. This would provide a more comprehensive solution for businesses seeking to optimize consumer behavior prediction and precision marketing.

## Author contributions

**Writing – original draft:** JIN LIN.

**Writing – review & editing:** JIN LIN.

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
