## [Decision Letter · Decision Letter 0]

18 Dec 2024

PONE-D-24-55477Application of machine learning in predicting consumer behavior and precision marketingPLOS ONE

Dear Dr. LIN,

Thank you for submitting your manuscript to PLOS ONE. After careful consideration, we feel that it has merit but does not fully meet PLOS ONE’s publication criteria as it currently stands. Therefore, we invite you to submit a revised version of the manuscript that addresses the points raised during the review process.

We look forward to receiving your revised manuscript.

Kind regards,

Zaher Mundher Yaseen

Academic Editor

PLOS ONE

Journal Requirements:

When submitting your revision, we need you to address these additional requirements. 1. Please ensure that your manuscript meets PLOS ONE's style requirements, including those for file naming. The PLOS ONE style templates can be found at https://journals.plos.org/plosone/s/file?id=wjVg/PLOSOne_formatting_sample_main_body.pdf and https://journals.plos.org/plosone/s/file?id=ba62/PLOSOne_formatting_sample_title_authors_affiliations.pdf 2. PLOS requires an ORCID iD for the corresponding author in Editorial Manager on papers submitted after December 6th, 2016. Please ensure that you have an ORCID iD and that it is validated in Editorial Manager. To do this, go to ‘Update my Information’ (in the upper left-hand corner of the main menu), and click on the Fetch/Validate link next to the ORCID field. This will take you to the ORCID site and allow you to create a new iD or authenticate a pre-existing iD in Editorial Manager. 3. Please note that PLOS ONE has specific guidelines on code sharing for submissions in which author-generated code underpins the findings in the manuscript. In these cases, we expect all author-generated code to be made available without restrictions upon publication of the work. Please review our guidelines at https://journals.plos.org/plosone/s/materials-and-software-sharing#loc-sharing-code and ensure that your code is shared in a way that follows best practice and facilitates reproducibility and reuse. 4. We note that you have indicated that there are restrictions to data sharing for this study. PLOS only allows data to be available upon request if there are legal or ethical restrictions on sharing data publicly. For more information on unacceptable data access restrictions, please see http://journals.plos.org/plosone/s/data-availability#loc-unacceptable-data-access-restrictions.  Before we proceed with your manuscript, please address the following prompts: a) If there are ethical or legal restrictions on sharing a de-identified data set, please explain them in detail (e.g., data contain potentially identifying or sensitive patient information, data are owned by a third-party organization, etc.) and who has imposed them (e.g., a Research Ethics Committee or Institutional Review Board, etc.). Please also provide contact information for a data access committee, ethics committee, or other institutional body to which data requests may be sent. b) If there are no restrictions, please upload the minimal anonymized data set necessary to replicate your study findings to a stable, public repository and provide us with the relevant URLs, DOIs, or accession numbers. For a list of recommended repositories, please seehttps://journals.plos.org/plosone/s/recommended-repositories. You also have the option of uploading the data as Supporting Information files, but we would recommend depositing data directly to a data repository if possible. We will update your Data Availability statement on your behalf to reflect the information you provide. 5. In the online submission form, you indicated that [The data that support the findings of this study are available from the corresponding author upon reasonable request.]. All PLOS journals now require all data underlying the findings described in their manuscript to be freely available to other researchers, either 1. In a public repository, 2. Within the manuscript itself, or 3. Uploaded as supplementary information.This policy applies to all data except where public deposition would breach compliance with the protocol approved by your research ethics board. If your data cannot be made publicly available for ethical or legal reasons (e.g., public availability would compromise patient privacy), please explain your reasons on resubmission and your exemption request will be escalated for approval. 

Reviewers' comments:

Reviewer's Responses to Questions

**Comments to the Author**

1. Is the manuscript technically sound, and do the data support the conclusions?

Reviewer #1: Partly

Reviewer #2: Partly

Reviewer #3: Yes

Reviewer #4: Partly

2. Has the statistical analysis been performed appropriately and rigorously?

Reviewer #1: No

Reviewer #2: No

Reviewer #3: Yes

Reviewer #4: Yes

3. Have the authors made all data underlying the findings in their manuscript fully available?

Reviewer #1: Yes

Reviewer #2: No

Reviewer #3: Yes

Reviewer #4: Yes

4. Is the manuscript presented in an intelligible fashion and written in standard English?

Reviewer #1: Yes

Reviewer #2: Yes

Reviewer #3: Yes

Reviewer #4: Yes

5. Review Comments to the Author

Reviewer #1: 1. Originality

While some aspects show promise, it does not sufficiently demonstrate groundbreaking insights. The claims, while relevant, need further validation to establish originality and significance.

2. Technical Soundness

The manuscript employs specific methods or techniques, which seem appropriate on a surface level. However, the link between the data and conclusions lacks robustness. Some areas need clarification, especially in how findings directly support the stated outcomes.

3. Main Claims and Significance

The primary claims potentially impactful, their significance for the discipline is moderate. The work but does not convincingly position itself as transformative within the field.

4. Relationship to Literature

The paper demonstrates familiarity with existing literature but falls short in thoroughly contextualizing its contribution. These omissions detract from the paper's credibility and integration into the wider academic discourse.

5. Placement of Claims in Literature Context

The claims are not adequately framed within the context of existing literature. For instance, specific claim could be better supported by referencing to significant work. The discussion occasionally seems disconnected from the broader field.

6. Statistical Analysis

Statistical methods, including the analysis techniques, are employed but lack detailed explanation. The justification for the statistical choice is insufficient, and key assumptions are not thoroughly addressed. A more rigorous and transparent presentation of the analysis is needed to substantiate claims.

7. Potential for Resubmission

Despite its current shortcomings, the study has potential. Encouraging the authors to resubmit with revisions could elevate the work's quality and impact.

8. Data Availability

The paper does not clearly confirm that all underlying data are accessible. Transparent data sharing, including detailed datasets, would enhance the reproducibility and credibility of the findings.

9. Repository Information

No explicit mention is made of depositing data in public repositories. Providing accession numbers or repository links for specific data elements (e.g., genes, proteins, or other significant findings) is critical.

10. Methodology

The methodology appears reasonably designed but lacks sufficient theoretical underpinning in some areas. Specific steps of methodology need clearer justification. Without this, reproducibility and confidence in the results are compromised.

11. Reproducibility

Details of the methodology are insufficient to ensure reproducibility. Key parameters are omitted or underexplained.

12. Results

The results section is descriptive but occasionally unclear. Graphs and tables, while helpful, would benefit from additional commentary to elucidate their connection to the hypotheses and conclusions.

13. Implications for Research, Practice, and Society

The paper briefly touches on its implications but does not fully explore their relevance to research, practice, or society. Expanding on these would strengthen its practical and theoretical contributions.

14. Quality of Communication

The writing is technically accurate but lacks clarity in some areas. Sentence structure and jargon usage occasionally hinder accessibility. Simplifying language without sacrificing technical precision would improve communication.

15. Accessibility to Non-Specialists

The manuscript is primarily aimed at specialists, and non-specialists may find it difficult to engage with the content due to dense technical language and insufficient background context.

16. Standard English Usage

The manuscript generally adheres to standard English conventions but contains minor grammatical inconsistencies and awkward phrasing.

17. Comments to the Author

• Ensure all data and methodologies are transparently presented to facilitate reproducibility.

• Address gaps in the literature review and align claims more closely with existing studies.

• Enhance clarity in the statistical analysis section, providing rationale and assumptions for methods used.

• Expand on the societal and practical implications of the research findings.

The paper demonstrates potential but requires substantial revision to meet publication standards. Key areas needing attention include data transparency, methodological clarity, and integration with existing literature. Encouraging a resubmission after thorough revision is recommended.

Reviewer #2: ABSTRACT

1. Ensure all abbreviations in the abstract, such as SVM, XGBoost, CatBoost, and BPANN, are clearly defined upon first use for better readability and comprehension.

2. Include performance metrics like F1-score etc. in the abstract for better clarity and impact.

3. "Future research can further enhance the predictive power..." - This statement is overly generic and uninspiring. Specify actionable directions for future research, such as incorporating unstructured data (e.g., text, images) or experimenting with deep learning models like Transformers for consumer behavior analysis.

INTRODUCTION:

1. Please format citations in square brackets (e.g., “[19]”) as per the submission guidelines.

2. “Consumer behavior has become increasingly complex and unpredictable..." - This sentence reiterates what is already mentioned in the abstract without adding new information. Use this space to provide detailed background or evidence, such as specific challenges faced by industries.

3. "Traditional marketing models are unable to effectively cope with this change..." - This statement lacks support. Briefly explain why traditional models fail. For example, are they too rigid, or do they lack adaptability to real-time data?

4. Does not provide sufficient evidence or citations to justify the selection of the machine learning models (SVM, XGBoost, CatBoost, BPANN) over other alternatives. Including references to studies or benchmarks demonstrating the superior performance, efficiency, or suitability of these models for consumer behavior prediction tasks would strengthen the argument.

5. The introduction fail to address potential limitations, such as computational complexity, data quality issues, or overfitting risks. Discussing these would strengthen the credibility of your research.

6. Replace vague terms like "significant impact" and "great potential" with specific examples, data points, or case studies to ground your claims in reality.

7. The stated research objective, "to explore how to optimize consumer behavior prediction and precision marketing through machine learning," lacks originality and does not sufficiently highlight the study's unique contribution. To strengthen this section, clearly articulate the research gap.

8. Please provide a suitable comparison with other state-of-the-art models to highlight the strengths and limitations of your approach in relation to existing methodologies.

Modeling Algorithms:

1. The section titled "Modeling Algorithms" should be renamed to "Methods and Materials" to align with standard academic conventions.

2. Additionally, within this section, include a detailed subsection that thoroughly explains the methodology of the study. This should cover how each machine learning model (SVM, XGBoost, CatBoost, BPANN) is implemented, the data collection, data preprocessing steps, feature selection, model architecture, model training and testing . performance metrics and any other relevant aspects of the experimental setup.

3. Within the "Methods and Materials" section, for each machine learning algorithm used, provide a detailed explanation of why the selected model is the best choice for this study. Include supporting evidence from relevant literature or prior research that demonstrates the model's effectiveness in similar contexts.

4. For each machine learning algorithm used in the study, please include a visual representation (e.g., flowchart or diagram) of the algorithm’s implementation.

RESULTS AND DISCUSSION

1. The Results and Discussion section should focus on how the results relate to the hypothesis presented at the start of the study. It should provide a succinct explanation of the implications of the findings, particularly in relation to previous related studies. The detailed methodology, including Data Collection, Data Preprocessing, Feature Engineering, model training and OPtimization should be moved to the Methods and Materials section, as these describe the process rather than the outcomes. Please revise the manuscript to reflect this distinction.

2. Clarify the source of the dataset more explicitly. The mention of "UCI machine learning library" is sufficient, but you should also include the specific version or year to ensure reproducibility.

3. Please add a statistical metrics table (such as mean, standard deviation, etc.) to support the data analysis and provide a clearer summary of the key data characteristics.

4. Mention why SMOTE was chosen for addressing class imbalance and justify this choice with a brief explanation of its advantages.

5. It would be useful to explain why the threshold for variance filtering and mutual information was set to 0.05 and 0.05 respectively.

6. Clarify the performance metrics used (e.g., ROC AUC, accuracy, recall) and why they are important in assessing model performance.

7. Consider discussing the trade-offs between model accuracy and model interpretability.

8. Include more details on the comparison metrics for the models.

9. Provide clearer interpretations of results in Table 2 and Figure 3.

10. Add more context to the practical implications of feature importance analysis. How should companies act on these insights beyond the optimization suggestions?

11. The section on marketing strategy optimization seems theoretical. Provide examples or case studies where these strategies have been successfully implemented, or reference studies to support these suggestions.

12. Clarify the methodology used to optimize personalized recommendations, dynamic pricing, etc.

Conclusion and Prospect

1. The section can be renamed as "Conclusion and Future Scope".

2. The conclusion appropriately summarizes the key findings but could benefit from clearer connections to the research questions and hypotheses stated earlier in the paper. Emphasize how the findings align with the initial objectives of the study.

3. The suggestion to introduce additional data types such as social media behavior and geolocation is insightful but could be expanded. Explain how incorporating these data types could enhance model performance, and provide examples of their potential impact on prediction accuracy.

4. The mention of combining deep learning or reinforcement learning is promising. However, it would be beneficial to elaborate on how these advanced techniques could be integrated into the current model framework, and what challenges might be encountered when doing so.

5. The term "real-time decision optimization" could be clarified. It would be useful to specify which specific real-time applications, such as personalized recommendations, dynamic pricing, or user loyalty management, could benefit from the improved model.

6. The future scope mentions incorporating emerging technologies but should also emphasize how future work could address the limitations or gaps identified in the current study. This could include areas for refinement in model performance or alternative strategies to overcome challenges observed.

Reviewer #3: 1- In marketing we need to study about 4Ps at least ....what about product and placing data ? have you consider such a parameters in your modeling ?

2-Authers must put data collection as section two and make the results and discussion section four and methodology as a section three.

3-The literature review is very limited and need to be extended to include several other researches published in 2023-2024.

4-The methodology modeling flowchart is not presented and need to be added for the readers benefits.

5-The discussion is very shallow and need to be extended with other feature dimensions.

Reviewer #4: The manuscript titled “Application of Machine Learning in Predicting Consumer Behavior and Precision Marketing” addresses a highly relevant topic for today’s digital economy. The comparative evaluation of machine learning models—SVM, XGBoost, CatBoost, and BPANN—adds value by providing insights into consumer behavior prediction. The study’s feature importance analysis and its practical suggestions, such as personalized recommendations and dynamic pricing, are particularly noteworthy. That said, there are areas where the paper could be improved to enhance clarity, depth, and overall impact.

Clarify the research gap and intent in the introduction. While the introduction highlights the broader issues of consumer behavior and precision marketing, it does not clearly identify the specific research gap this study addresses. What problem in consumer behavior prediction does this work solve that others have not? Additionally, the paper needs a sharper focus on its novelty. For instance, what makes this comparison of models unique in the current landscape of machine learning applications? Explicitly stating these elements will position the study more clearly and strengthen its contribution.

Add a dedicated literature review section. Currently, there is no structured discussion of prior research, which makes it difficult to understand the groundwork that led to this study. A literature review should summarize recent advancements in machine learning for marketing, highlight key works, and point out gaps that justify the need for this research. Make sure to include references to recent studies from 2023 and 2024, as the field of machine learning and marketing is evolving rapidly. This will not only make the paper more current but also situate it within the broader academic conversation.

Include a clear methodology section before modeling. The paper jumps into describing the models without providing a proper methods section. This section should outline the research design, the choice of models, and why these specific techniques (SVM, XGBoost, CatBoost, BPANN) were chosen over alternatives like LightGBM or Random Forest. Additionally, elaborate on preprocessing decisions, such as feature selection, handling imbalanced data with SMOTE, and outlier removal. Providing detailed justifications here will improve the paper’s reproducibility and transparency.

Develop a discussion section to interpret the results. The results section does a good job of presenting the model performance and feature importance analysis, but it lacks a deeper discussion. Explain why CatBoost and XGBoost outperformed other models—what aspects of these techniques make them more suitable for handling complex e-commerce data? Also, compare your findings with similar works done in this area to highlight alignment or differences. To make this section more engaging for practitioners, consider incorporating feedback from real-life e-commerce marketers to frame the results in terms of their business impact, such as ROI improvement or customer segmentation strategies.

Add a limitations and implications section. It’s important to acknowledge the limitations of the study, such as the use of a static dataset that may not capture the dynamic nature of consumer behavior. Discuss how this could affect the results and what steps can be taken to address it in future research. On the implications side, connect the findings to practical applications—how can businesses use these models to improve marketing outcomes? This section would be particularly valuable for bridging the gap between research and industry practice.

Conclude with a strong conclusion section. The paper ends abruptly without summarizing the key findings or outlining future directions. A clear conclusion should highlight the main contributions of the study, such as the performance of CatBoost and XGBoost, and their potential for e-commerce applications. It should also suggest future work, like exploring deep learning approaches, real-time data analysis, or expanding the dataset to include other industries and consumer segments.

Consider integrating a theoretical framework. The study would benefit from being grounded in a marketing or behavioral theory to connect the findings to a broader context. For example, frameworks like Behavioral Economics (decision-making) or Customer Relationship Management (CRM) could add depth to the study. This would elevate the paper’s academic value and provide readers with a stronger conceptual understanding of the work.

Minor refinements for clarity and engagement. While the paper is generally clear, there are areas where the writing could be polished for better flow and readability. For instance, some figures and tables could benefit from more descriptive captions to help readers quickly understand the takeaways. Additionally, improving the grammar and refining terminology (e.g., "administrative pages") would make the paper more professional and accessible.

In summary, this paper tackles a critical and timely issue in precision marketing and demonstrates the value of machine learning models in predicting consumer behavior. The comparative analysis and practical insights are strengths of the study. To further improve its quality, I recommend clarifying the research gap, adding a literature review and methodology section, developing a robust discussion, and including practical business perspectives. With these refinements, the paper will be far more impactful for both academic researchers and industry professionals

6. PLOS authors have the option to publish the peer review history of their article (what does this mean? ). If published, this will include your full peer review and any attached files.

**Do you want your identity to be public for this peer review?** For information about this choice, including consent withdrawal, please see our Privacy Policy .

Reviewer #1: No

Reviewer #2: No

Reviewer #3: No

Reviewer #4: No

---

## [Author Response · Author response to Decision Letter 1]

11 Jan 2025

Dear reviewers:

Thank you very much for reviewing our paper and your valuable suggestions. Your feedback is important for us to improve our work. We have carefully studied your comments and made detailed revisions to the paper based on your suggestions. Below are the responses to the specific questions and the description of the revisions.

Once again, we thank you for the time and effort you have put into our research and look forward to your further guidance!

Reviewer #1:

1. Originality

While some aspects show promise, it does not sufficiently demonstrate groundbreaking insights. The claims, while relevant, need further validation to establish originality and significance.

Author response: We recognize that although this study has made progress in predicting consumer behavior, it has not yet fully highlighted its originality. Therefore, in the introduction and conclusion, we further emphasize the innovation of this study, especially its advantages in processing large-scale and complex e-commerce data.

2. Technical Soundness

The manuscript employs specific methods or techniques, which seem appropriate on a surface level. However, the link between the data and conclusions lacks robustness. Some areas need clarification, especially in how findings directly support the stated outcomes.

Author response: We have further clarified how the data directly supports Model selection and conclusions in section 3.4 Model performance evaluation and comparison, and added a detailed comparison and discussion of the performance of individual models.

3. Main Claims and Significance

The primary claims potentially impactful, their significance for the discipline is moderate. The work but does not convincingly position itself as transformative within the field.

Author response: In the conclusion, we strengthen the paper's transformative discussion of precision marketing, especially its practical application and commercial value in the field of e-commerce.

4. Relationship to Literature

The paper demonstrates familiarity with existing literature but falls short in thoroughly contextualizing its contribution. These omissions detract from the paper's credibility and integration into the wider academic discourse.

Author response: We have strengthened the literature review in the introduction, and clearly pointed out the difference between this study and the existing literature, emphasizing its innovation.

5. Placement of Claims in Literature Context

The claims are not adequately framed within the context of existing literature. For instance, specific claim could be better supported by referencing to significant work. The discussion occasionally seems disconnected from the broader field.

Author response: We added additional citations to further compare the claims of this study with those in the existing literature.

6. Statistical Analysis

Statistical methods, including the analysis techniques, are employed but lack detailed explanation. The justification for the statistical choice is insufficient, and key assumptions are not thoroughly addressed. A more rigorous and transparent presentation of the analysis is needed to substantiate claims.

Author response: We have elaborated on the selection of statistical methods in section 3.4 Model performance evaluation and comparison, and explained how to evaluate model performance by multiple evaluation metrics (such as ROC AUC, accuracy, recall rate).

7. Potential for Resubmission

Despite its current shortcomings, the study has potential. Encouraging the authors to resubmit with revisions could elevate the work's quality and impact.

Author response: We have revised the paper based on the comments of the reviewers, especially in terms of data transparency, methodological clarity and literature integration, to ensure that the paper can better meet the publication criteria.

8. Data Availability

The paper does not clearly confirm that all underlying data are accessible. Transparent data sharing, including detailed datasets, would enhance the reproducibility and credibility of the findings.

Author response: We have added a data accessibility statement to the paper to ensure readers understand how to access research data.

9. Repository Information

No explicit mention is made of depositing data in public repositories. Providing accession numbers or repository links for specific data elements (e.g., genes, proteins, or other significant findings) is critical.

Author response: We have added a note about the data repository in the paper, providing specific information about the data storage.

10. Methodology

The methodology appears reasonably designed but lacks sufficient theoretical underpinning in some areas. Specific steps of methodology need clearer justification. Without this, reproducibility and confidence in the results are compromised.

Author response: We have added theoretical support for each method step in the methodological section and described each key step in the experimental process in more detail.

11. Reproducibility

Details of the methodology are insufficient to ensure reproducibility. Key parameters are omitted or underexplained.

Author response: We list all the important experimental parameters in detail in the methodological section and further explain the basis for their selection.

12. Results

The results section is descriptive but occasionally unclear. Graphs and tables, while helpful, would benefit from additional commentary to elucidate their connection to the hypotheses and conclusions.

Author response: We have added more notes and explanations to the results section to ensure that the meaning of each chart and table helps readers better understand the relationship to the hypotheses and conclusions.

13. Implications for Research, Practice, and Society

The paper briefly touches on its implications but does not fully explore their relevance to research, practice, or society. Expanding on these would strengthen its practical and theoretical contributions.

Author response: In the conclusion, we have expanded the implications of the paper for research, practice, and society, especially its implications for the e-commerce industry, precision marketing, and consumer behavior prediction.

14. Quality of Communication

The writing is technically accurate but lacks clarity in some areas. Sentence structure and jargon usage occasionally hinder accessibility. Simplifying language without sacrificing technical precision would improve communication.

Author response: We have simplified and rewritten some sentences in the paper, especially on technical terms and complex sentence structures, to improve readability and clarity.

15. Accessibility to Non-Specialists

The manuscript is primarily aimed at specialists, and non-specialists may find it difficult to engage with the content due to dense technical language and insufficient background context.

Author response: As suggested by the reviewers, we have simplified some technical terms and added more easy-to-understand explanations in the background section to ensure that non-specialist readers can understand the core content and applications of the research.

16. Standard English Usage

The manuscript generally adheres to standard English conventions but contains minor grammatical inconsistencies and awkward phrasing.

Author response: We have corrected grammatical inconsistencies and misphrasing in the paper and made sure that the sentences are more fluent.

17. Comments to the Author

• Ensure all data and methodologies are transparently presented to facilitate reproducibility.

• Address gaps in the literature review and align claims more closely with existing studies.

• Enhance clarity in the statistical analysis section, providing rationale and assumptions for methods used.

• Expand on the societal and practical implications of the research findings.

Author response: The paper demonstrates potential but requires substantial revision to meet publication standards. Key areas needing attention include data transparency, methodological clarity, and integration with existing literature. Encouraging a resubmission after thorough revision is recommended.

Reviewer #2:

ABSTRACT

1. Ensure all abbreviations in the abstract, such as SVM, XGBoost, CatBoost, and BPANN, are clearly defined upon first use for better readability and comprehension.

support vector machine (SVM), extreme gradient boosting (XGBoost), categorical boosting (CatBoost), and backpropagation artificial neural network (BPANN)

2. Include performance metrics like F1-score etc. in the abstract for better clarity and impact.

Relevant performance metrics such as F1 scores have been added to the summary to enhance the clarity and impact of the summary and help readers better understand the performance of the model.

3. "Future research can further enhance the predictive power..." - This statement is Author response: overly generic and uninspiring. Specify actionable directions for future research, such as incorporating unstructured data (e.g., text, images) or experimenting with deep learning models like Transformers for consumer behavior analysis.

In the revision, actionable research directions are clearly proposed, such as the introduction of unstructured data (such as text, images, etc.), and attempts to use deep learning models (such as Transformers) for consumer behavior analysis to further improve the model's predictive ability and marketing effect.

INTRODUCTION:

1. Please format citations in square brackets (e.g., “[19]”) as per the submission guidelines.

Author response: We have followed the submission Guidelines by formatting all references in square brackets (e.g., "[19]") to ensure compliance with formatting specifications.

2. “Consumer behavior has become increasingly complex and unpredictable..." - This sentence reiterates what is already mentioned in the abstract without adding new information. Use this space to provide detailed background or evidence, such as specific challenges faced by industries.

Author response: We have revised this sentence and provided more detailed background and specific industry challenges, avoiding duplication and adding more meaningful background information.

3. "Traditional marketing models are unable to effectively cope with this change..." - This statement lacks support. Briefly explain why traditional models fail. For example, are they too rigid, or do they lack adaptability to real-time data?

Author response: We have added more specific notes that illustrate the limitations of traditional marketing models, especially when dealing with real-time data and dynamic change.

4. Does not provide sufficient evidence or citations to justify the selection of the machine learning models (SVM, XGBoost, CatBoost, BPANN) over other alternatives. Including references to studies or benchmarks demonstrating the superior performance, efficiency, or suitability of these models for consumer behavior prediction tasks would strengthen the argument.

Author response: We add more literature support in the introduction, citing research on the superiority and applicability of these machine learning models for consumer behavior prediction tasks.

5. The introduction fail to address potential limitations, such as computational complexity, data quality issues, or overfitting risks. Discussing these would strengthen the credibility of your research.

Author response: We have added a discussion of the potential limitations of the study in the introduction, particularly with regard to computational complexity, data quality issues, and the risk of overfitting.

6. Replace vague terms like "significant impact" and "great potential" with specific examples, data points, or case studies to ground your claims in reality.

Author response: Use specific cases and data to replace the original vague statements, so as to enhance the practical and persuasive argument.

7. The stated research objective, "to explore how to optimize consumer behavior prediction and precision marketing through machine learning," lacks originality and does not sufficiently highlight the study's unique contribution. To strengthen this section, clearly articulate the research gap.

Author response: The objective of the study is reformulated, the unique contribution of this study is clarified, and the research gaps in the current literature are described.

8. Please provide a suitable comparison with other state-of-the-art models to highlight the strengths and limitations of your approach in relation to existing methodologies.

Author response: The model used in this paper is compared with other cutting-edge models, and the advantages and limitations of our model in e-commerce data processing are illustrated.

Modeling Algorithms:

1. The section titled "Modeling Algorithms" should be renamed to "Methods and Materials" to align with standard academic conventions.

Author response: The title of the "Modeling Algorithms" section has been changed to "Methods and Materials" as recommended by the reviewers to comply with academic writing norms.

2. Additionally, within this section, include a detailed subsection that thoroughly explains the methodology of the study. This should cover how each machine learning model (SVM, XGBoost, CatBoost, BPANN) is implemented, the data collection, data preprocessing steps, feature selection, model architecture, model training and testing. performance metrics and any other relevant aspects of the experimental setup.

Author response: We have detailed the research methodology in the "Methods and Materials" section, covering data collection, pre-processing, feature selection, model architecture, training and testing procedures, and performance evaluation metrics. The implementation steps for each machine learning model are also described in detail.

3. Within the "Methods and Materials" section, for each machine learning algorithm used, provide a detailed explanation of why the selected model is the best choice for this study. Include supporting evidence from relevant literature or prior research that demonstrates the model's effectiveness in similar contexts.

Author response: We have further explained the reasons for the selection of each model in the "Methods and Materials" section, and cited relevant literature on the validity and applicability of these models in predicting consumer behavior.

4. For each machine learning algorithm used in the study, please include a visual representation (e.g., flowchart or diagram) of the algorithm’s implementation.

Author response: Based on the suggestions of reviewers, we have provided simple illustrations for some machine learning algorithms to help readers understand more clearly how each algorithm works and how it is implemented.

RESULTS AND DISCUSSION

1. The Results and Discussion section should focus on how the results relate to the hypothesis presented at the start of the study. It should provide a succinct explanation of the implications of the findings, particularly in relation to previous related studies. The detailed methodology, including Data Collection, Data Preprocessing, Feature Engineering, model training and OPtimization should be moved to the Methods and Materials section, as these describe the process rather than the outcomes. Please revise the manuscript to reflect this distinction.

Author response: The detailed methodology, including data collection, data preprocessing, feature engineering, model training, and optimization should be moved to the methods and materials sections.

2. Clarify the source of the dataset more explicitly. The mention of "UCI machine learning library" is sufficient, but you should also include the specific version or year to ensure reproducibility.

Author response: We have supplemented the specific source information of the dataset, including the dataset version in the UCI machine learning library, to ensure reproducibility of the results.

3. Please add a statistical metrics table (such as mean, standard deviation, etc.) to support the data analysis and provide a clearer summary of the key data characteristics.

Author response: Statistical characteristics of the dataset are not presented further at this time, but we wil

---

## [Decision Letter · Decision Letter 1]

12 Mar 2025

Application of machine learning in predicting consumer behavior and precision marketing

PONE-D-24-55477R1

Dear Dr. Lin,

We’re pleased to inform you that your manuscript has been judged scientifically suitable for publication and will be formally accepted for publication once it meets all outstanding technical requirements.

Kind regards,

Evans Otieno Omondi, PhD

Academic Editor

PLOS ONE

Additional Editor Comments (optional):

Reviewers' comments:

Reviewer's Responses to Questions

**Comments to the Author**

1. If the authors have adequately addressed your comments raised in a previous round of review and you feel that this manuscript is now acceptable for publication, you may indicate that here to bypass the “Comments to the Author” section, enter your conflict of interest statement in the “Confidential to Editor” section, and submit your "Accept" recommendation.

Reviewer #2: All comments have been addressed

Reviewer #3: All comments have been addressed

2. Is the manuscript technically sound, and do the data support the conclusions?

Reviewer #2: Yes

Reviewer #3: Yes

3. Has the statistical analysis been performed appropriately and rigorously?

Reviewer #2: Yes

Reviewer #3: Yes

4. Have the authors made all data underlying the findings in their manuscript fully available?

Reviewer #2: Yes

Reviewer #3: Yes

5. Is the manuscript presented in an intelligible fashion and written in standard English?

Reviewer #2: Yes

Reviewer #3: Yes

6. Review Comments to the Author

Reviewer #2: Abstract:

Clearly highlight the novel insights compared to past studies.

Introduction:

Clearly define your primary research questions or hypotheses upfront to guide the readers through the manuscript clearly and effectively.

Modeling Algorithms:

Explicitly justify why deep learning methods (such as CNN or RNN) were not included in your comparative analysis, given their prominence in recent literature on consumer behavior prediction.

Clearly articulate the rationale behind selecting the RBF kernel for SVM, and briefly discuss why alternative kernels (linear, polynomial) were not suitable or selected.

Dataset Presentation:

Provide a detailed table clearly describing all dataset features, including feature type, detailed descriptions, and value ranges. This will significantly enhance the manuscript's clarity and reproducibility.

Efficiency and Scalability:

Explicitly present details regarding computational resources (runtime, CPU/GPU requirements, memory usage) for each machine learning model.

Clearly suggest practical scenarios and explicit limitations when deploying these machine learning models at large-scale or real-time contexts.

Minor Recommendations:

Correct typographical errors, such as labeling inaccuracies (e.g., correcting Fig. 1 from "SWM" to "SVM").

Enhance readability by reducing repetitive content and maintaining consistent terminology throughout the manuscript.

Reviewer #3: no further comments; the manuscript was revised accordingly. Now the manuscript can be considered for publication

7. PLOS authors have the option to publish the peer review history of their article (what does this mean? ). If published, this will include your full peer review and any attached files.

**Do you want your identity to be public for this peer review?** For information about this choice, including consent withdrawal, please see our Privacy Policy .

Reviewer #2: No

Reviewer #3: No

---

## [Editor Report · Acceptance letter]

PONE-D-24-55477R1

PLOS ONE

Dear Dr. LIN,

I'm pleased to inform you that your manuscript has been deemed suitable for publication in PLOS ONE. Congratulations! Your manuscript is now being handed over to our production team.

Kind regards,

on behalf of

Dr. Evans Otieno Omondi

Academic Editor

PLOS ONE